# An Integrated Sample Referral System for HIV Viral Load and Early Infant Diagnosis in North-Western Province, Zambia—A Retrospective Cross-Sectional Study

**DOI:** 10.3390/healthcare12060618

**Published:** 2024-03-08

**Authors:** Doreen Mainza Shempela, Jay Sikalima, Jim Mwandia, Ernest Mwila, Rodgers Chilyabanyama, Mike Masona, Cynthia Banda Kasonde, Andrew Mwandila, Hector Kamalamba, Maisa Kasanga, Imukusi Mutanekelwa, Steward Mudenda, Franco Jordan Kandama, Fatim Cham, Michael Njuguna, Paul McCarrick, Linden Morrison, Victor Daka, Karen Sichinga

**Affiliations:** 1Churches Health Association of Zambia, Lusaka 10101, Zambia; jay.sikalima@chaz.org.zm (J.S.); jim.mwandia@chaz.org.zm (J.M.); ernest.mwila@chaz.org.zm (E.M.); rodgers.chilyabanyama@chaz.org.zm (R.C.); mike.masona@chaz.org.zm (M.M.); cynthia.kasonde@chaz.org.zm (C.B.K.); andrew.mwandila@chaz.org.zm (A.M.); jordankandama1993@gmail.com (F.J.K.); karen.sichinga@chaz.org.zm (K.S.); 2Laboratory Department, Solwezi General Hospital, Solwezi 10101, Zambia; hectorkamalamba@gmail.com; 3Department of Epidemiology and Biostatistics, School of Public Health, Zhengzhou University, Zhengzhou 450001, China; kasangaanita@gmail.com; 4Public Health Department, Michael Chilufya Sata School of Medicine, Copperbelt University, Ndola 21692, Zambia; kusi789@gmail.com (I.M.); dakavictorm@gmail.com (V.D.); 5Department of Pharmacy, School of Health Sciences, University of Zambia, Lusaka 10101, Zambia; freshsteward@gmail.com; 6Global Fund to Fight AIDS, Tuberculosis and Malaria (GFATM), 1218 Geneva, Switzerland; fatim.jallow@theglobalfund.org (F.C.); michael.njuguna@theglobalfund.org (M.N.); paul.mccarrick@theglobalfund.org (P.M.); linden.morrison@theglobalfund.org (L.M.)

**Keywords:** HIV viral load, coverage, turnaround time, sample referral system, Zambia

## Abstract

Zambia’s adult HIV prevalence is high at 11% and faces challenges in achieving UNAIDS 95-95-95 targets for HIV, with a national viral load suppression of 86.2% falling short of the required 95%. North-Western Province has the lowest viral load suppression at 77.5%. Our study investigated the role of an integrated sample referral system in optimizing HIV viral load coverage and Early Infant Diagnosis turnaround time in the province. Using electronic data from the DISA Laboratory Information System and Smartcare, a retrospective cross-sectional analysis was conducted, involving 160,922 viral load and Early Infant Diagnosis results. The chi-square test and multiple linear regression were used for analysis. Following the implementation of the integrated sample referral system, viral load coverage consistently increased monthly (*p* < 0.001), Early Infant Diagnosis turnaround time improved by 47.7%, and sample volume increased by 25%. The study identifies associations between various factors and testing outcomes. These findings demonstrate improvements in viral load coverage and the Early Infant Diagnosis turnaround time and suggest targeting modifiable factors to further optimize the referral system. We recommend continued strengthening of the referral system and more deliberate demand-creation implementation strategies.

## 1. Introduction

HIV prevalence in Zambia among individuals aged 15 years and above continues to remain high at 11% [1]. HIV infection has a profound impact on communities, as many people living with HIV (PLHIV) are parents and caregivers that are supposed to attend to the needs of their families [2,3]. Some of the greatest challenges of PLHIV are stigma and discrimination, with both known to affect the quality of life and treatment outcomes negatively. The Joint United Nations Program on HIV/AIDS (UNAIDS) 95-95-95 targets requires that 95% of individuals should know their HIV status, and among those who know their status, 95% should be on treatment, while 95% on treatment should reach viral load suppression [4]. Zambia has met the first and second 95 of the UNAIDS 95-95-95 and is at 86.2% for the national viral load (VL) suppression for the third 95 [1]. In the absence of a suitable vaccine candidate or cure for people living with HIV (PLHIV), key secondary prevention modalities including routine HIV VL and Early Infant Diagnosis (EID) monitoring to track HIV risk reduction in addition to ART adherence are key [5].

Health system strengthening is a key driver of improved health outcomes among PLHIV in Zambia. The infrastructure supporting HIV VL and EID testing has been recognized by the WHO as a priority for system strengthening and monitoring the progress toward 95-95-95 UN goals [6]. This should be supported by a robust health information system for HIV such as the DISA LAB LIS [6,7]. A total of 15 hospitals in Zambia have viral load machines supported by both the government of Zambia and the private sector. In North-Western Province (NWP) of Zambia, HIV VL testing is done at Solwezi General Hospital in the Solwezi district, the provincial capital of NWP [8]. The linkage of these testing platforms to complement and avail testing to the target population is paramount [9]. The Ministry of Health of Zambia through support from the Churches Health Association of Zambia (CHAZ) is involved in the last-mile transportation of HIV VL and EID samples from primary and secondary healthcare facilities to the provincial testing site via a spoke and hub model [10]. Given this background, this study examined the contribution of the integrated sample referral system for optimizing HIV VL coverage and EID testing in NWP in Zambia.

## 2. Materials and Methods

### 2.1. Study Site and Overview of Referral System

North-Western Province (NWP) is one of the ten provinces in Zambia and has an area of 126,386 square km and a provincial capital located in Solwezi. NWP has 11 districts, a combined population of 1,270,028 and the highest annual growth rate at 4.8% [11]. The referral system in NWP serves 11 districts and over 327 healthcare facilities ranging from second-level facilities (Mission hospitals and District hospitals) to primary healthcare facilities (Health centers and Health posts). The flow of samples and information is from the requesting peripheral health facilities using motorbikes through the 14 hub facilities based in the districts via the spoke and hub model (Figure 1) [10]. There are 14 hub facilities supported by CHAZ. The CHAZ-supported VL referral system in NWP started in January 2022. Of the 14 hub facilities, 8 have DISA LAB LIS installed and functional, while the remaining 6 do not. Hence, the dissemination of electronic results from the testing lab to the 8 hub facilities is via the DISA LAB LIS and hardcopy for the remaining 6 hub facilities. Using the spoke and hub model (see Figure 1), the hubs then ensure that hardcopy results are delivered to all requesting health facilities where connectivity via internet, electricity, and printing are usually a major challenge [10].

Solwezi General Hospital started offering HIV VL testing in late 2015 and was available for other sites within the province to send samples for processing. DISA LAB LIS was introduced at SGH in October 2017 to facilitate continuity of reporting and integration with other provinces. CHAZ has supported the critical sample referral capital infrastructure in NWP, including m-PIMA Point of Care analyzers, transportation (3 motor vehicles, 30 motorbikes), VL storage facilities (cooler boxes at each facility, 10 mobile portable freezers), and information storage/dissemination (computers and printers in all 6 hub facilities). Some program consumables include fuel, lab reagents, internet subscriptions, and stationery, while expenses involved include overhead costs, motor vehicle servicing, and allowances that assist in ensuring uninterrupted service [10]. The final laboratory products from this comprehensive investment are VL results originating from the DISA LAB LIS on soft or hard copy. The results are then fed into Smartcare (Plus version 16.6) at the facility level by dedicated persons handling data management and entry. Smartcare is an electronic health records management software that houses clinical information taken from history, physical examination, and ordering of various diagnostics for monitoring PLHIV on ART. Records in Smartcare are entered in real time during a clinical visit or later on if the system is down. Smartcare, adopted as the national standard electronic health records software, has been in use since 2010 and is deployed in all provinces, most districts, and most health facilities countrywide. The flow of information for Smartcare is the same as in DISA, with databases from facilities merged at the district level, and district databases merged at the provincial level all the way up to the national level [12]. Using de-identified reports from Smartcare, VL testing coverage can be assessed by comparing the annual number of ART patients with a documented VL result in the clinical or laboratory records (TX_PVLS_D) with the total number of PLHIV on ART (TX_CURR) six months prior.

### 2.2. Study Type

This was a retrospective review of data on the integrated sample referral system in North-Western Province. This study included all healthcare facilities in NWP with PLHIV on treatment and those which collect viral load and EID samples and take them to Solwezi General Hospital for testing. The data collected included HIV VL or EID results for any individual collected within NWP from June 2021 to October 2023. All healthcare facilities that use Smartcare in the province were included in the study. HIV VL or EID samples and Smartcare data from other provinces outside NWP and rejected samples were excluded from the present study.

### 2.3. Sample Size Determination

This was a complete enumeration of VL and EID electronic data from four out of eight randomly selected hub facilities with functional DISA LAB LIS installed. Randomization was achieved using the Microsoft Excel “RANDBETWEEN” function. The eight hub facilities were assigned whole numbers from 1 to 8, and Microsoft Excel produced four random numbers corresponding to the selected four hub facilities. A complete enumeration of provincial data from Smartcare for the study period was used.

### 2.4. Data Collection

Data from June 2021 to October 2023 were accessed in the DISA LAB LIS system at four CHAZ-supported hub facilities, namely Chavuma Mission Hospital in Chavuma district, Mukinge Mission Hospital in Kasempa district, Kalene Mission Hospital in Ikelenge district, and Loloma Mission Hospital in Manyinga district. The electronic data from DISA LAB LIS were exported into Excel. The accession report and the viral load results report were exported and later merged. Information entered in the DISA LAB LIS was from routine validated national lab requisition forms. The information collected included the date of sample collection, date of receipt in the lab, date of accession, date of results review, test type, sample type, facility name, indications for collection (routine or targeted), and lab result. The Smartcare aggregated database for North-Western Province was accessed for the report on number of ART patients with a documented VL result in the medical or laboratory records in the previous 12 months (TX_PVLS_D) and the total number of PLHIV on ART (TX_CURR) in the previous six months. The data for the study period June 2021 to October 2023 were exported from Smartcare.

### 2.5. Data Analysis

The primary objective was to explore changes in VL testing coverage before and after the implementation of the integrated samples referral system. Hence, to demonstrate the impact of the integrated referral system, data collected from Smartcare were analyzed using multiple linear regression with Newey–West standard errors and were presented using an interrupted time series. The initial baseline VL testing coverage of 53.1% in January 2022 at the commencement of the integrated samples referral system was noted. All data from Smartcare were analyzed using STATA version 16.0. The secondary objective was to determine the turnaround time (TAT), which is categorical and was measured in days. TAT was defined as the time interval from the receipt of the sample by the testing lab (at Solwezi General Hospital) to the result being verified and printed for collection. TAT overall was defined as the time interval from the collection of the sample by the peripheral health facility through the districts to the testing lab (at Solwezi General Hospital) and to the time the result reached the requesting health facility. TAT before and after the introduction of the CHAZ-supported VL sample referral system was also compared. The chi-square test was used for analysis after satisfying all assumptions, otherwise, Fisher’s Exact Test was used. A two-tailed *p*-value of 0.05 was used for statistical significance. Subgroup comparisons were done with post hoc analysis, and Bonferroni adjustment *p*-values were used. Bonferroni adjustment *p*-values were determined by dividing the alpha (0.05) by the number of comparisons on the dependent variable. Data from DISA LAB LIS were analyzed using Microsoft Excel (Version 2016, Microsoft, Redmond, WA, USA) and SPSS version 26.

### 2.6. Ethical Consideration

Permission to carry out the study was sought from the Provincial Health Office for NWP and the selected hub facilities. Data collected from the DISA LAB LIS and Smartcare was de-identified to ensure privacy and confidentiality. The data were stored in a password-protected computer with access restricted to only the research team. Ethical clearance was obtained from the Tropical Diseases Research Centre Ethics Review Board (IRB number 00002911, approval date: 4 December 2023).

## 3. Results

### 3.1. Distribution of People Living with HIV

A total of 160,922 viral load and Early Infant Diagnosis results exported from the DISA LAB LIS were reviewed. Demographics showed that most (73.9%) people living with HIV were in the age group of 15–49 years. A few (10.3%) were below 15 years, and the rest (15.8%) were above 49 years. In addition, most (66.1%) people living with HIV were female, see Table 1.

### 3.2. Effect of the Integrated Samples Referral System on VL Testing Coverage

The analysis revealed a downward viral load coverage trend of 0.0002% per month. Notably, in the inaugural month of the intervention, a decrease of 0.0055% in viral load coverage occurred. However, this decline was transient, as it was succeeded by a statistically significant surge of 0.001% (*p* < 0.001, CI [0.0006–0.0014]). Further bolstering this finding is the lincom estimate, derived by specifying the post trend. This estimate showed that following the introduction of the integrated sample referral system, viral load coverage demonstrated a consistent average monthly increase at a rate of 0.0008% (*p* < 0.001, CI = [0.0006–0.0010]), see Figure 2 and Appendix A.

The TAT at Solwezi General Hospital for both VL and EID between January 2020 and November 2023 was less than 7 days for 78.6% of the samples processed. Less than a quarter (21.4%) of the TAT was above 7 days. After the introduction of the integrated referral system, the largest change in the TAT at SGH was seen for results received within 3 days with a 9% increment (Figure 3).

### 3.3. Turnaround Time at SGH

Figure 4 shows the overall TAT between 2020 to 2023 between SGH and peripheral facilities. The overall TAT was defined as the time interval from the collection of the sample by the peripheral health facility through the districts to the testing lab (at Solwezi General Hospital) and the result reaching the requesting health facility, shown in Figure 4 above. The turnaround time for both VL and EID between 2020 and 2023 was mostly (76.9%) within 14 days. Only a few (23.1%) lab results reached the facilities after 14 days. After the introduction of the integrated sample referral system, the largest improvement (9.2%) in the overall turnaround time was noted with results received within 4 to 7 days.

### 3.4. Viral Load and Early Infant Diagnosis at Solwezi General Hospital

Figure 5 below shows the contribution of each test to the TAT at Solwezi General Hospital (SGH) before and after the introduction of the integrated sample referral system. A 2.3% increment in VL TAT within 7 days and an increment of 47.7% in EID TAT within 3 days was noted at SGH after the introduction of the integrated sample referral system.

### 3.5. Sample Volume

The volume of VL and EID samples sent to SGH for processing increased by 25.3%, or 18,068, after the introduction of the integrated referral system, see Figure 6. The integrated sample referral system was designed to improve both quantity and quality.

### 3.6. Association between TAT and Other Variables

The chi-square test was used to determine relationships between TAT and other predictor variables. The analysis showed the overall TAT was significantly affected by year (*p* < 0.0005), sample referral intervention (*p* < 0.0005), age (*p* < 0.0005), district (*p* < 0.0005), test ordered (*p* < 0.0005), and sample type (*p* < 0.0005), see Table 2.

Post hoc comparisons showed that the year 2020 experienced statistically significant higher overall TAT of 8–14 days and >14 days compared to those of years 2022 and 2023. The relationship between overall TAT and age group showed a significant relationship, with individuals aged <15 years and overall TAT of ≤3 days, 4–7 days, 8–14 days, and >14 days. A similar relationship pattern was noted in the age group >49 years with a significant relationship with overall TAT of ≤3 days, 4–7 days, and 8–14 days. Despite the distance, the top four furthest districts from Solwezi district reported significant TAT times as follows: the Chavuma district reported results availability within 7 days, Ikelenge reported between 4 and 14 days, Zambezi reported within 14 days, while Mwinilunga reported 4–7 days and >14 days. Facilities within Solwezi received results within 14 days. The rest of the districts nearest to Solwezi showed variability in overall TAT ranging from shorter to longer times. Regarding the test ordered and the overall TAT, subgroup analysis revealed that VL and EID test results were received at any time, whether within 3 days, between 4 to 7 days, 8 to 14 days, or after 14 days. Sample type and test type were correlated as DBS is used for EID and plasma for VL, hence the subgroup analysis of sample type showed the same relationship with TAT as test ordered, see Table 2.

## 4. Discussion

The primary objective of the study was to explore changes in VL testing coverage before and after implementation of the integrated sample referral system, and the secondary objective was to determine the turnaround time (TAT) for VL and EID in North-Western Province, Zambia. Our study population showed that the HIV positivity rate was twice as high among women compared to that in men. These findings corroborate findings from another study in South Africa where the prevalence was higher in women compared to that in men [13]. Another study which aggregated data from 1990 to 2016 found that prevalence in females was almost twice as high in females as in men [14]. This is in contrast to HIV rates in Europe, where the disease is more concentrated in men, with men having sex with men as the main key population [15]. The analysis unveiled insights into the referral system’s impact on viral load coverage in the CHAZ program in North-Western Province. The observed initial downward trend before the intervention signaled a potential problem, prompting timely action. Despite an initial viral load coverage decline in the first intervention month, fluctuations in early implementation are typical [16,17]. Subsequent months showed a significant increase in viral load coverage, indicating a positive response to the intervention and emphasizing the system’s meaningful impact. Studies in Zambia and South Africa support the notion that an efficient sample transport system can substantially enhance viral load coverage from 10% to 90% [16,18]. However, additional interventions, such as improved turnaround time, quality improvement, and effective data management, may complement the referral system for optimal viral load coverage [18,19]. The lincom estimate revealed a consistent monthly 0.0008% increase in viral load coverage, reinforcing the referral system’s effectiveness. This finding is similar to those of other studies that suggest a multiple intervention approach to improve the outcomes of HIV treatment [17,18,20]. The positive findings suggest the system’s contribution to sustained improvement, emphasizing logistic interventions’ significance in timely sample transportation. While the immediate impact is promising, ongoing monitoring and real-time data-based adjustments are crucial for long-term program optimization.

The TAT at Solwezi General Hospital for Viral Load (VL) and Early Infant Diagnosis (EID) mostly adhered to 7 days, while overall TAT generally stayed within 14 days. Despite TAT improvements over the years, laboratory testing still presents delays compared to point-of-care testing, impacting treatment options and clinical outcomes for patients, especially those with high viral loads. Further reductions in TAT at SGH and overall TAT can be achieved by addressing challenges such as like machine breakdowns. The study’s TAT aligns with that in a Tanzanian retrospective cohort study, indicating comparability in TAT from phlebotomy to peripheral facilities and patient communication [21]. The time when results are available at the peripheral site up to the time they are viewed for clinical decision making was not explored in this study, and this can be looked at in a separate mixed-methods study.

Despite difficulties in comparing TAT across jurisdictions due to differences in TAT definitions, each laboratory in the country must establish their own means of acceptable TAT based on clinical needs and users’ input and be in line with recommendations from the International Organisation for Standardisation (ISO) guidelines [22]. The local acceptable TAT at SGH for VL is 7 days, and 14 days at the national level. In addition, the local hospital policy is that with one available PCR machine, 80% of samples received should be processed within 7 days, and with two PCR machines, this is increased to 95%. The results for the TAT at SGH for VL were within the acceptable range for both national and local hospital policy using one PCR machine.

The EID TAT witnessed a significant improvement, shifting from being largely outside the national standard to aligning with both local and national acceptable norms of 3 days and 7 days, respectively. However, further efforts are required to reduce EID TAT, as some results exceeded the locally acceptable threshold of 3 days. The extended TAT for EID is attributed to transportation delays, particularly in districts outside Solwezi. Despite prolonged EID TAT at SGH, hub facilities have GeneXpert and m-PIMA machines primarily reserved for point-of-care testing for EID and pregnant mothers to offset delays. The innovative use of point-of-care machines aligns with a Kenyan study associating TAT with viral suppression among adolescents and young adults [23]. Most studies reviewed focused on VL TAT unlike EID TAT, hence this study is not comparable with others.

Before the introduction of the referral system, the results showed that both tests (VL and EID) were equally prioritized in terms of reducing the overall TAT, and this also correlated with the DBS sample used for EID and the plasma sample for VL testing. In 2017, the Ministry of Health supported the deployment of point-of-care (POC) EID/VL and the use of diagnostic network optimization (DNO) to improve facility testing and demand creation [24]. This assisted in reducing the demand for sample couriers in centralized testing. Due to the COVID-19 pandemic, which brought in competing demands and stress on the health system, most campaign programs had to realign to support the pandemic efforts, leaving VL and EID services struggling. After the introduction of the integrated referral system, priority was given to EID samples leading to an almost 50% increment in EID TAT within 3 days. This was within both the local and national acceptable EID TAT of within 3 days and 7 days, respectively. Prioritizing EID is a major step towards achieving the prevention of mother-to-child transmission during the breastfeeding stage and ultimately helps with HIV epidemic control by reducing the number of new infections.

The year 2020, before the introduction of the integrated referral system, showed that higher TAT was noted compared to that of the other years. The reduction in TAT over the years can be partially attributed to the introduction of the CHAZ-supported referral system. The reduction in TAT after the introduction of an additional referral system resonates with a Nigeria study that used third-party logistics to improve the timely availability of results and ultimately patient care [25].

The Coronavirus disease 2019 (COVID-19) was first reported in Wuhan City, China in 2019 and spread to countries worldwide, resulting in the World Health Organization (WHO) declaring it a pandemic [26]. This led to massive disruptions in health systems globally [27]. The implementation of the integrated courier system was at the height of the COVID-19 pandemic. During the pandemic, there was a higher priority given to the testing of COVID-19 samples. The testing for COVID leveraged the courier in the testing system due to the centralized nature of testing at Solwezi General Hospital. This approach has been reported to be effective in pandemic situations such as with COVID-19 [27]. This could have helped reduce the TAT for HIV VL and EID as COVID-19 samples were considered urgent as the pandemic unfolded. The advent of the COVID-19 pandemic also necessitated the integration of testing on already established VL and EID platforms, possibly increasing the TAT for VL and EID. The challenges in the scale up of VL and EID amidst the COVID-19 pandemic have been documented elsewhere [28]. Other challenges could have included staffing challenges with some staff falling ill from COVID-19 and others being reassigned to attend COVID-19 testing. Additionally, the anxieties that were brought about by the COVID-19 pandemic could have led to poor health seeking behaviors and thus to a lower uptake of VL and EID services, as described in a previous study [29].

Despite some districts being the furthest (Chavuma, Zambezi) or with the poorest road network (Ikelenge, Mwinilunga) from Solwezi district, they reported lower TAT times compared to that of some facilities nearer to Solwezi district. A major reason for this is that the DISA LAB LIS is installed in those facilities to facilitate electronic, real-time transfer of lab results from Solwezi immediately after they are available. In addition, the hub facilities with DISA LAB LIS participate in accessioning and inputting the patient details into DISA LAB LIS before the sample reaches Solwezi, and this reduces the processing time. Longer distance to the testing site, unavailability of transport, and poor transport network are three common reasons for longer TAT as reported by various authors in Malawi, Kenya, and Lesotho [22,30,31]. These challenges are equally faced in North-Western Province because it has the second largest area (126,386 square km) [11] offering a vast geographic road network, longer distances to peripheral facilities, and unique geospatial challenges such as complete cut-off by road for some facilities above the Zambezi River during rainy season due to flooding and connectivity problems during the rainy season.

This study also noted that a few DBS samples were used for adults as an alternative sample type for testing viral load and/or genotype resistance. Compared to blood or plasma, DBS offers unique advantages of transport and storage and is a WHO-recommended good substitute in resource-limited settings or when extensive transport challenges exist, such as in faraway districts [32].

The volume of samples processed at SGH highlighted an increase after the introduction of the integrated referral system. Before the introduction of the referral systems, a major challenge was samples spending a longer than normal duration at the peripheral health facility, and this potentially demoralized staff from collecting samples. Authors have reported degenerative changes affecting the HIV VL for blood/plasma samples stored beyond the recommended period, which is usually between 2 and 8 °C for 5 days [31]. The introduction of the referral system, laboratory staff ferrying samples in the peripheral facilities to ensure quality, and community sensitization of key stakeholders in the districts were responsible for the increased numbers, and this demonstrates increased demand creation for routine VL and EID testing. Demand creation interventions can be implemented at the community/client level, provider level, and facility or systems level. More demand creation can be done to further improve the VL and EID TAT over the years in the province. Community-focused demand creation has been demonstrated in various jurisdictions across the globe for its effectiveness at increasing the usage of routine VL testing [32].

## 5. Limitations

Routinely collected and retrospective data are not well suited for research purposes. Missing data entries were noted, and this was handled during the statistical analysis. Despite the notable benefits brought about by the integrated sample courier system, we also noted that the challenges brought about by the COVID-19 could have affected the optimization of the courier system. An evaluation in a stable post-pandemic era is imperative.

## 6. Conclusions

The study demonstrates significant and consistent improvements in HIV VL testing coverage during the study period. Furthermore, VL TAT within local and national acceptable ranges of 7 days and 14 days, respectively, and a major improvement in EID TAT to within acceptable national standards of 7 days after the introduction of the integrated referral system was noted. Improvements in sample volume suggest increased uptake; however, to further improve the coverage and TAT, there is a need to explore more VL demand creation opportunities. Our findings show that deliberate sample courier initiatives can significantly contribute to access to VL and EID services. This experience warrants recommending the strengthening of sample courier systems to enable robust laboratory networks to support VL and EID testing.

## Figures and Tables

**Figure 1 healthcare-12-00618-f001:**
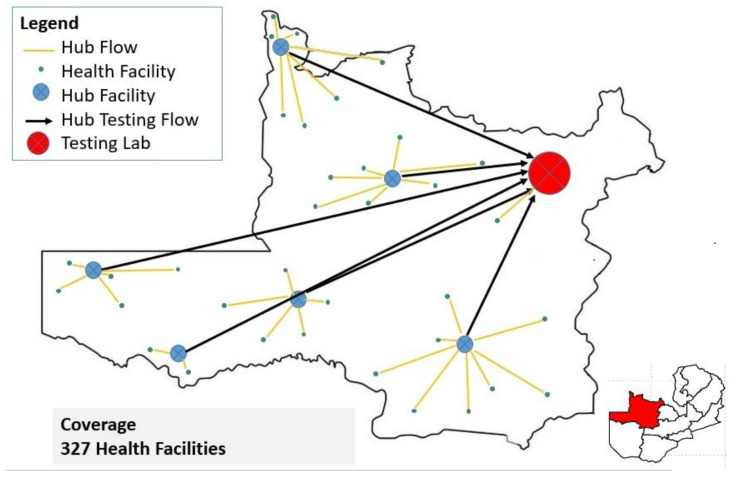
The spoke and hub referral network for North-Western Province.

**Figure 2 healthcare-12-00618-f002:**
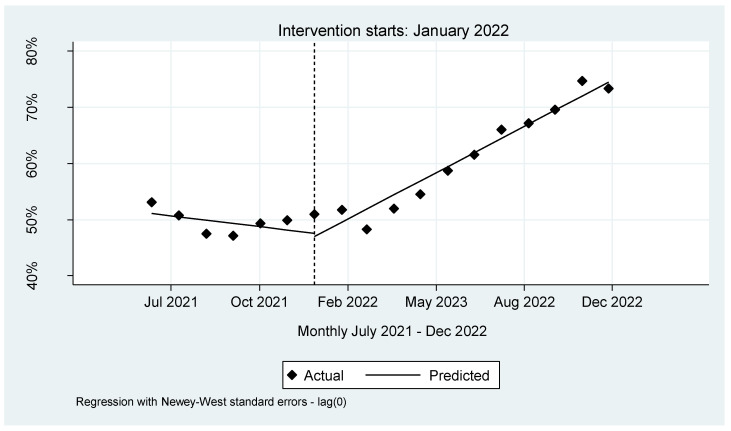
Interrupted time series for VL coverage in NW province.

**Figure 3 healthcare-12-00618-f003:**
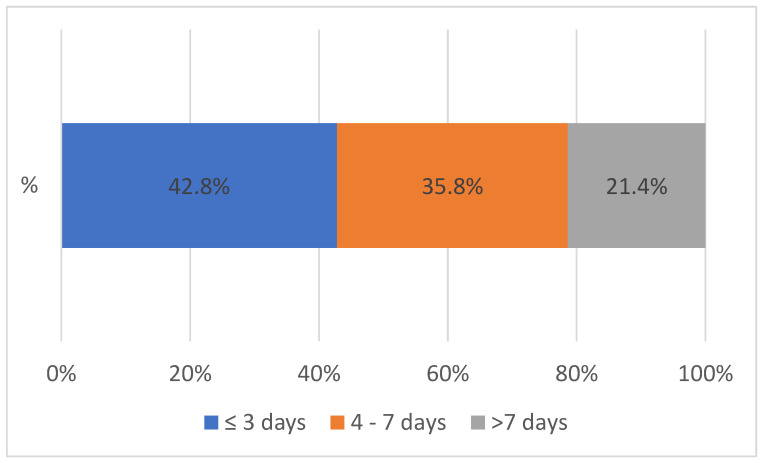
Turnaround time at Solwezi General Hospital between 2020 and 2023, n = 91,387.

**Figure 4 healthcare-12-00618-f004:**
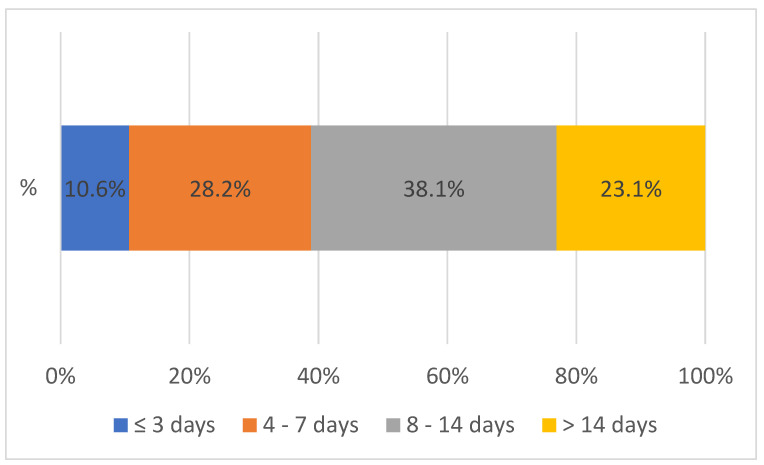
Overall turnaround time between 2020 and 2023 from the peripheral facility to SGH and vice versa, n = 72,700.

**Figure 5 healthcare-12-00618-f005:**
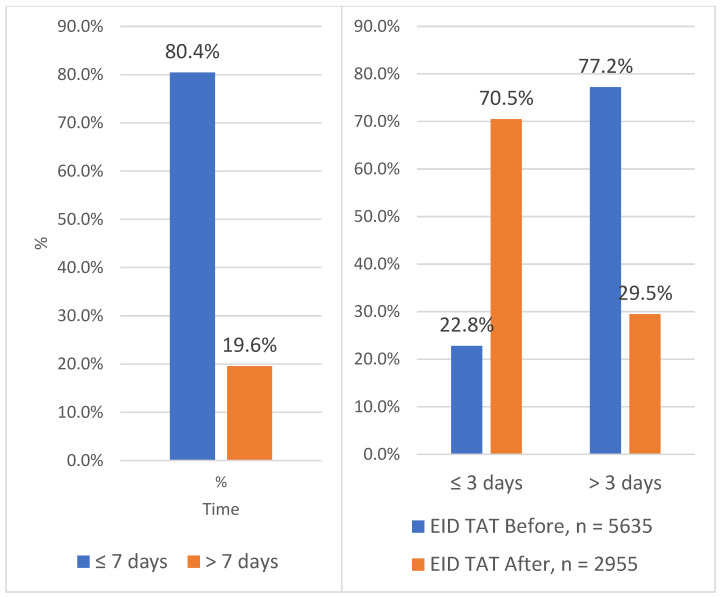
Turnaround time for EID and VL at Solwezi General Hospital between 2020 and 2023.

**Figure 6 healthcare-12-00618-f006:**
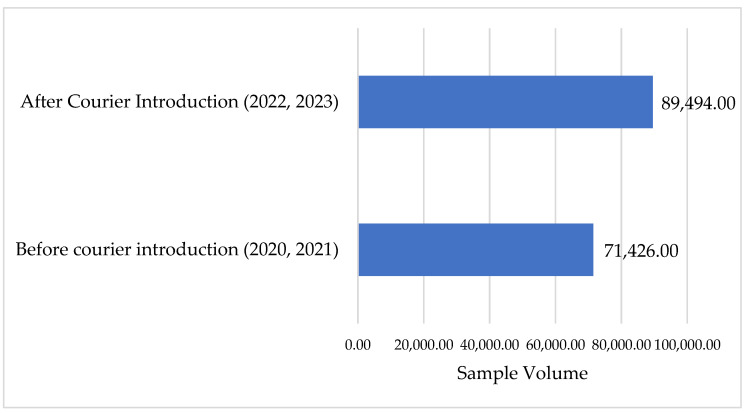
Sample volume at SGH before and after the introduction of the referral system.

**Table 1 healthcare-12-00618-t001:** Distribution of people living with HIV.

Variable	Frequency	%
Age, n = 156,350		
<15 years	16,073	10.3
15–49 years	115,518	73.9
>49 years	24,759	15.8
Gender, n = 158,184		
Male	53,690	33.9
Female	104,494	66.1

n = number of participants.

**Table 2 healthcare-12-00618-t002:** Factors associated with the overall TAT from 2020 to 2023.

Variable	TAT Overall					*p* Value
	Totals, n (%) *	≤3 Days	4–7 Days	8–14 Days	>14 Days	
Intervention, n = 72,696 Before Referral (2020/21) After Referral (2022/23)	36,839 (100) 35,857 (100)	4245 (11.5) 3486 (9.7)	8713 (23.7) 11,807 (32.9)	15,015 (40.8) 12,662 (35.3)	8866 (24.1) 7902 (22.0)	0.001
Year, n = 72,700 2020 2021 2022 2023	17,781 (100) 19,059 (100) 18,717 (100) 17,143 (100)	1834 (10.3) 2411 (12.7) 2380 (12.7) 1106 (6.5)	5012 (28.2) 3701 (19.4) 5735 (30.6) 6075 (35.4)	8156 (45.9) 6859 (36.0) 6221 (33.2) 6441 (37.6)	2779 (15.6) 6088 (31.9) 4381 (23.4) 3521 (20.5)	0.001
Age, n = 71,256 <15 years 15–49 years >49 years	6639 (100) 54,095 (100) 10,522 (100)	552 (8.3) 5728 (10.6) 1259 (12.0)	1655 (24.9) 15,303 (28.3) 3181 (30.2)	2379 (35.8) 21,042 (38.9) 3776 (35.9)	2053 (30.9) 12,022 (22.2) 2306 (21.9)	0.001
District, n = 72,480 Chavuma (650 km) Ikelenge (397 km) Kabompo (360 km) Kalumbila (150 km) Kasempa (186 km) Manyinga (360 km) Mufumbwe (250 km) Mushindano (90 km) Mwinilunga (286 km) Solwezi (reference) Zambezi (520 km)	1069 (100) 620 (100) 2650 (100) 6818 (100) 2811 (100) 965 (100) 3643 (100) 3381 (100) 3849 (100) 42,868 (100) 3806 (100)	195 (18.2) 63 (10.2) 529 (20.0) 516 (7.6) 485 (17.3) 70 (7.3) 289 (7.9) 250 (7.4) 371 (9.6) 4402 (10.3) 507 (13.3)	207 (19.4) 108 (17.4) 1226 (46.3) 1418 (20.8) 1212 (43.1) 264 (27.4) 1586 (43.5) 767 (22.7) 1358 (35.3) 10,881 (25.4) 1445 (38.0)	381 (35.6) 302 (48.7) 559 (21.1) 2828 (41.5) 632 (22.5) 339 (35.1) 1061 (29.1) 1358 (40.2) 1415 (36.8) 17,855 (41.7) 900 (23.6)	286 (26.8) 147 (23.7) 336 (12.7) 2056 (30.2) 482 (17.1) 292 (30.3) 707 (19.4) 1006 (29.8) 705 (18.3) 9730 (22.7) 954 (25.1)	0.001
Test, n = 72,693 Viral Load Early Infant Diagnosis	69,002 (100) 3691 (100)	7546 (10.9) 185 (5.0)	19,756 (28.6) 766 (20.8)	26,392 (38.2) 1284 (34.8)	15,308 (22.2) 1456 (39.4)	0.001
Sample type, n = 72,408 Blood Dry Blood Spot Plasma	6312 (100) 6912 (100) 59,184 (100)	648 (10.3) 367 (5.3) 6673 (11.3)	1739 (27.6) 1469 (21.3) 17,236 (29.1)	3303 (52.3) 2752 (39.8) 21,508 (36.3)	622 (9.9) 2324 (33.6) 13,767 (23.3)	0.001

* = row percent, n = number of participants, TAT—turnaround time, km—kilometers.

## Data Availability

Data are contained within the article.

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
