# Peer review of "An Integrated Sample Referral System for HIV Viral Load and Early Infant Diagnosis in North-Western Province, Zambia—A Retrospective Cross-Sectional Study"

_healthcare, 2024, doi:10.3390/healthcare12060618_

Round 1
Reviewer 1 Report
Comments and Suggestions for Authors
Thank you for allowing me to review this manuscript. This manuscript titled “An Integrated Sample Referral system for HIV Viral Load and 2 Early infant diagnosis in North Western Province, Zambia – a 3 retrospective Cross-sectional study” is an interesting article of great relevance today, although it has several limitations. which make it suitable for publication in this Journal.
These limitations are detailed below:
- In the introduction it would be important to add more quotes that justify the importance of the topic of study, bringing the author closer to the context of the study. It would be interesting to address the importance of the impact that people diagnosed with HIV have on their daily lives, which makes addressing this disease essential, compared to others with less social stigma and physical and psychological impact. Some articles that could complete this statement:
- The material and methods section does not reflect whether the sample is representative and how the sample calculation was carried out.
- Results section: The acronyms used in the tables are not always specified at the bottom of the table.
- In the conclusions, due to the great importance of the topic today, it would be interesting to include a future line and the implications of the present study in clinical practice.
good job
Reviewer 2 Report
Comments and Suggestions for Authors
I express my gratitude for the opportunity to critically review the manuscript titled "An Integrated Sample Referral System for HIV Viral Load and Early Infant Diagnosis in North Western Province, Zambia– A Retrospective Cross-Sectional Study." As implied by its title, the researchers investigated the impact of a referral system on the early diagnosis of children living with HIV, a highly relevant topic not only in Zambia, where the infection prevalence is considerably high. The manuscript is well-written and presents data that will be of interest to the journal. I offer the following considerations to the authors.
Major Comments:
Comment 1: The study design is before-and-after. While the presented evidence appears convincing regarding the impact of the referral system on the event of interest, and the authors extensively discuss structural and logistical aspects of the local healthcare system, it is imperative to also discuss other contextual factors that could have influenced the observed outcome. For instance, were there campaigns promoting mass screenings in the target population? Were there educational interventions by the government or other non-governmental organizations? Addressing these aspects is crucial to bolster the presented results.
Minor Comments:
Comment 2: In Table 1, it is noted that the majority of individuals living with HIV are women (approximately 2 out of every 3). It is worth briefly discussing this, as in other regions, particularly in the Western world, infections tend to be skewed towards particularly young men.
Comment 3: Also concerning Table 1, is it possible to further stratify the age groups, especially those under 15 years? The prevalence is high (10.3%), and as a public health professional, it is important for me to identify children who may have been infected through vertical transmission versus those whose risk factor was sexual contact. Therefore, I suggest at least identifying those children analyzed who were 12 months or younger at the time of diagnosis.
Comment 4: The analyzed information primarily corresponds to the data generated during the emergency phase of the COVID-19 pandemic (June 2021 to October 2023). It is worthwhile to discuss, if applicable and if the authors have relevant data, the impact of the pandemic, such as the burden on health services, on the diagnostic opportunity for HIV infection.
Round 2
Reviewer 2 Report
Comments and Suggestions for Authors
I would like to thank the authors for taking the time to review the comments and suggestions that I previously provided. I believe that the manuscript has improved substantially, however my suggestion regarding the analysis of the effect of the pandemic on the event of interest remains valid. The authors mentioned that this is not the objective of the study, but given the significant change observed in the utilization of health services, I believe that the effect of the pandemic should at least be discussed.
Among many other aspects, during the emergency phase of the pandemic, health resources were redirected to care for patients with COVID-19. Additionally, patients without respiratory symptoms were afraid to go to hospitals for fear of being infected. These and many other factors could be related to the observations during the last period of the analysis.
